



# Reduction in $C_2H_6$ from 2015 to 2020 over Hefei, eastern China points to air quality improvement in China

Youwen Sun[1], Hao Yin[1,3,*], Cheng Liu[1,2,4,5,*], Emmanuel Mahieu[6], Justus Notholt[7], Yao Té[8], Xiao Lu[9], Mathias Palm[7], Wei Wang[1], Changong Shan[1], Qihou Hu[1], Min Qin[1], Yuan Tian[10], and Bo Zheng[11]

[1]*Key Laboratory of Environmental Optics and Technology, Anhui Institute of Optics and Fine Mechanics, HFIPS, Chinese Academy of Sciences, Hefei 230031, China*

[2]*Center for Excellence in Regional Atmospheric Environment, Institute of Urban Environment, Chinese Academy of Sciences, Xiamen, 361021, China*

[3]*University of Science and Technology of China, Hefei, 230026, China*

[4]*Key Laboratory of Precision Scientific Instrumentation of Anhui Higher Education Institutes, University of Science and Technology of China, Hefei, 230026, China*

[5]*Anhui Province Key Laboratory of Polar Environment and Global Change, USTC, Hefei, 230026, China*

[6]*Institute of Astrophysics and Geophysics, University of Liège, Belgium*

[7]*University of Bremen, Institute of Environmental Physics, P. O. Box 330440, 28334 Bremen, Germany*

[8]*Laboratoire d'Etudes du Rayonnement et de la Matière en Astrophysique et Atmosphères (LERMA-IPSL), Sorbonne Université, CNRS, Observatoire de Paris, PSL Université, 75005 Paris, France*

[9]*School of Engineering and Applied Sciences, Harvard University, Cambridge, MA 02138, USA*

[10]*Anhui University Institutes of Physical Science and Information Technology, Hefei 230601, China*

[11]*Institute of Environment and Ecology, Tsinghua Shenzhen International Graduate School, Tsinghua University, Shenzhen 518055, China*

*Corresponding authors.

E-mail addresses: chliu81@ustc.edu.cn (C. Liu), yhyh95@mail.ustc.edu.cn (H. Yin)

**Abstract**

Ethane ($C_2H_6$) is an important greenhouse (GHG) gas and plays a significant role in tropospheric chemistry and climate change. This study first presents and quantifies the variability, source, and transport of $C_2H_6$ over densely populated and industrialized eastern China by using ground-based high-resolution Fourier transform infrared (FTIR) remote sensing technique. We obtained a retrieval error of $6.21 \pm 1.2$ (1σ)% and degrees of freedom (DOFS) of $1.47 \pm 0.2$ (1σ) in retrieval of $C_2H_6$ tropospheric column-averaged dry-air mole fraction (troDMF) over Hefei, eastern China (117°E, 32°N, 30 m a.s.l.). The observed $C_2H_6$ troDMF reached a minimum monthly mean value of $(0.36 \pm 0.26)$ ppbv in July and a maximum monthly mean value of $(1.76 \pm 0.35)$ ppbv in December, and showed a negative change rate of $(-2.60 \pm 1.34)$ %/yr from 2015 to 2020. The dependencies of $C_2H_6$ troDMF on meteorological and emission factors were analyzed by using generalized additive models (GAMs). Generally, both meteorological and emission factors have positive influences on $C_2H_6$ troDMF in cold season (DJF/MAM) and negative influences on $C_2H_6$ troDMF in warm season (JJA/SON). GEOS-Chem chemical model simulation captured the





observed C₂H₆ troDMF variability and was thus used for source attribution. GEOS-Chem model
sensitivity simulations concluded that the anthropogenic emissions (fossil fuel plus biofuel
emissions) and the natural emissions (biomass burning plus biogenic emissions) accounted for 49.2%
and 37.1% of C₂H₆ troDMF abundance over Hefei, respectively. The observed C₂H₆ troDMF
abundance mainly results from the emissions within China (74.1%), where central, eastern, and
northern China dominated the contribution (57.6%). Seasonal variability in C₂H₆ transport inflow
and outflow over the observation site is largely related to the mid-latitude westerlies and Asian
monsoon system. Reduction in C₂H₆ abundance from 2015 to 2020 mainly results from the decrease
in local and transported C₂H₆ emissions, which points to air quality improvement in China in recent
years.
Keywords: Remote sensing; FTIR; Ethane; Climate change; GAMs
**1. Introduction**
Ethane (C₂H₆) is an important greenhouse (GHG) gas and one of the most abundant Volatile
Organic Compounds (VOCs) in the atmosphere (Abad et al. 2011; Singh et al. 2001; Steinfeld 1998).
Although C₂H₆ is much less abundant than methane (CH₄) and also less efficient relative to mass,
it plays a significant role in tropospheric chemistry and climate change (Tzompa-Sosa et al. 2017).
In the presence of nitrogen oxides (NOₓ = NO + NO₂), the C₂H₆ oxidation can enhance tropospheric
ozone (O₃) generation, which shows a positive radiative influence on climate (Sun et al. 2018a) and
threatens crop yields (Sun et al. 2018a; Van Dingenen et al. 2009) and human health (Sun et al.
2018a; Tzompa-Sosa et al. 2017). In addition, as a major source of acetaldehyde (CH₃CHO), C₂H₆
has a great impact on the production of peroxyacetyl nitrate (PAN) which is a key reservoir species
of NOₓ (Fischer et al. 2014). The main sink of tropospheric C₂H₆ is predominantly destruction via
reaction with the hydroxyl radical (OH)(Xiao et al. 2008), which determines the residence time of
most tropospheric species (Steinfeld 1998). As a result, tropospheric C₂H₆ can decrease the
atmospheric oxidative capacity and indirectly impact the climate by extending the CH₄ lifetime
(Monks et al. 2018; Taylor et al. 2020). Atmospheric C₂H₆ has a relatively long residence time of a
few months (Franco et al. 2016), allowing it to undergo intercontinental transport. As a result,
observations of C₂H₆ can be assimilated into a chemical transport model to estimate nonlocal
emissions and air quality, and provide valuable insights into model biases of C₂H₆ simulations
(Tzompa-Sosa et al. 2017).
On a global scale, the main sources of C₂H₆ are leakage from production, processing, and
transport of natural gas (62%), biofuel combustion (20%) and biomass burning emission (18%),
largely occurred in the Northern Hemisphere (NH) (Franco et al. 2016; Xiao et al. 2008). Additional
minor sources of C₂H₆ are from biogenic and oceanic sources. However, on a regional scale, the
proportion of each C₂H₆ source may show large difference. Natural gas leakage contribution can
reach 80% of C₂H₆ emissions in regions with active oil and natural gas production (Gilman et al.
2013), where C₂H₆ emissions are highly correlated with CH₄ emissions. In such regions, C₂H₆ can
be applied as a tracer for separation of fossil fuel CH₄ emissions from multiple methane (CH₄)
sources (e.g., oil and gas, cows, wetlands, and rice yield) (McKain et al. 2015; Roscioli et al. 2015).
The C₂H₆ abundance in the Southern Hemisphere (SH) is much lower than those in the NH since
the anthropogenic C₂H₆ sources are low in the SH, and the residence time of C₂H₆ is shorter than
the interhemispheric exchange rate. Many studies concluded that C₂H₆ in the SH is primarily emitted





from biomass burning, and is closely correlated with CO and HCN emissions (Notholt et al. 2000;
Rinsland et al. 2002; Vigouroux et al. 2012; Zeng et al. 2012).
$C_2H_6$ is one of the target gases of a global ground-based Fourier transform infrared
spectroscopy (FTIR) network, namely the Network for Detection of Atmospheric Composition
Change (NDACC) (De Mazière et al. 2018). FTIR time series of $C_2H_6$ with different time periods
have been reported in many stations for validation of satellite data or chemical model simulation
(Abad et al. 2011; Franco et al. 2015; Franco et al. 2016; Glatthor et al. 2009), or evaluation of local
air quality and air pollutants transport caused by anthropogenic emission and biomass burning
(Angelbratt et al. 2011; Lutsch et al. 2016; Lutsch et al. 2019; Nagahama and Suzuki 2007; Rinsland
et al. 2002; Simpson et al. 2012; Viatte et al. 2015; Viatte et al. 2014; Vigouroux et al. 2012; Zeng
et al. 2012; Zhao et al. 2002). Several FTIR sites have observed the decrease in $C_2H_6$ over 1990 –
2010, and characterized consistent interannual trends in the −1 to −2.7% $yr^{-1}$ range (Franco et al.
2015; Franco et al. 2016; Simpson et al. 2012; Zeng et al. 2012). This declining trend has been
largely attributed to the reduction of global fugitive emissions (Franco et al. 2015; Simpson et al.
2012). Recently, several studies concluded that the long-term decline in $C_2H_6$ in the NH reversed
from 2009 onwards (Franco et al. 2015; Franco et al. 2016). Using ground-based FTIR $C_2H_6$ total
columns derived at five selected NDACC sites, Franco et al. (2016) characterized the $C_2H_6$
evolution from 2009 – 2015 and determined growth rates of ∼3% $yr^{-1}$ at remote sites and of ∼5%
$yr^{-1}$ at mid-latitudes. This change is mainly attributed to the exploitation of shale gas and tight oil
reservoirs in North America (Franco et al. 2016; Helmig et al. 2016).
The NDACC network has been operating for almost three decades around the globe (De
Maziere et al. 2018; Sun et al. 2018a). However, most instruments are located in Europe and
Northern America, but the number of observation sites in the rest parts of world remains sparse, and
there is only one qualified observations site in China, i.e., the Hefei site (117°E, 32°N, 30 m a.s.l.)
located in a densely populated and industrialized area in eastern China (Sun et al. 2018a). The Hefei
site is not yet affiliated to the NDACC network but its observation routine is following the NDACC
standard convention since 2015 (Sun et al. 2018a). As the consequence of a series of actions for
emission control, air pollution over China in recent years has been significantly improved (Zhang
et al. 2019; Zheng et al. 2018). However, the atmospheric pollution over densely populated and
industrialized eastern China is still in severe situation (Zhang et al. 2019; Zheng et al. 2018). The
complexity, extension, and severity of the atmospheric pollution in eastern China are still unrivaled
compared to the rest of world (Lu et al. 2018; Zheng et al. 2018). FTIR observations at Hefei have
been used extensively for evaluation of satellite data (Tian et al. 2018; Wang et al. 2017), chemical
model simulation (Tian et al. 2018; Yin et al. 2020; Yin et al. 2019), local air quality (Shan et al.
2019; Sun et al. 2018a) and the transport of air pollutants caused by anthropogenic and biomass
burning emissions (Sun et al. 2018a; Sun et al. 2020a; Sun et al. 2020b).
In this study, we first present and quantify the variability, source, and transport of $C_2H_6$ over
densely populated and industrialized eastern China by using FTIR observation, GEOS-Chem model
simulation, and atmospheric circulation pattern techniques. The seasonality and interannual
variability of $C_2H_6$ over Hefei, eastern China from 2015 – 2020 are investigated. The dependencies
of $C_2H_6$ on meteorological and co-emitted gases (hereafter emission factors) are analyzed by using
generalized additive models (GAMs)(Wood and Simon 2004). The ground-based FTIR $C_2H_6$ time
series are for the first time applied to evaluate the GEOS-Chem model for the specifics of $C_2H_6$
simulation over eastern China. We further run a series of GEOS-Chem sensitivity simulations to





quantify relative contributions of various source categories and regions to the observed $C_2H_6$
abundance. The three-dimensional (3D) transport inflow and outflow pathways of $C_2H_6$ over the
observation site are finally determined by the GEOS-Chem sensitivity simulations and atmospheric
circulation pattern. This study can not only enhance the understanding of regional emission,
transport, and air clean actions over eastern China, but also contribute to form new reliable remote
sensing data in this sparsely-monitored regions for climate change research.
The next section describes the retrieval for FTIR tropospheric column-averaged dry-air mole
fraction (troDMF) of $C_2H_6$, the configuration of GEOS-Chem model simulation, and the GAMs
regression approach. Section 3 reports the variability of $C_2H_6$ troDMF and comparison with the
GEOS-Chem simulation. Section 4 reports the GAMs regression results and the interpretation.
Section 5 reports the results for source attribution using GEOS-Chem sensitivity simulation and
atmospheric circulation pattern. We conclude the study in Section 6.
**2. Methods and data**
**2.1 $C_2H_6$ troDMF retrieval**
The $C_2H_6$ troDMF time series were calculated by using direct solar absorption spectra saved
with a FTIR spectrometer in operation at Hefei, eastern China (Sun et al. 2018a; Tian et al. 2018).
Site description and instrumentation can be found in (Sun et al. 2018a). Briefly, the FTIR
observatory includes a high resolution FTIR spectrometer (IFS125HR, Bruker) and a solar tracker
(Solar Tracker-A 547, Bruker). This FTIR observatory alternately saved near infrared (NIR) and
middle infrared (MIR) solar spectra in routine observations (Tian et al. 2018). The NIR and MIR
spectra are saved with different spectral resolutions but both of them can be used to retrieve total
columns and volume mixing ratio (VMR) profiles of a variety of trace gases in the atmosphere. The
MIR spectra used in present work are saved with a spectral resolution of $0.005cm^{-1}$ following the
requirements of NDACC standard convention (http://www.ndacc.org/, last accessed on 27
December 2020). The FTIR instrument is equipped with a KBr beam splitter, a filter centered at
$2800\ cm^{-1}$, and an InSb detector for $C_2H_6$ measurements. The number of $C_2H_6$ measurements on
each measurement day varied from 1 to 17 with an average of 6. In total, there were 743 days of
qualified measurements between 2015 and 2020.
In this study, the VMR profile of $C_2H_6$ was first retrieved by using the SFIT4 algorithm updated
from SFIT2 (Pougatchev et al. 1995) and implementing the optimal estimation method (Rodgers
2000). The $C_2H_6$ troDMF was then calculated by taking a weighting average of the $C_2H_6$ VMR
profile and the air mass using a fixed tropospheric altitude. The $C_2H_6$ VMR profile was retrieved in
a broad window of $2976 - 2978\ cm^{-1}$. The VMR profiles of $CH_4$ and $H_2O$ and column of $O_3$ were
also retrieved together with the $C_2H_6$ VMR profile for minimizing the atmospheric absorption
interference. Spectroscopic absorption parameters of all gases are based on the atm16 line list from
the compilation of Geoffrey Toon (Sun et al. 2020a). The *a priori* vertical profiles of temperature
$H_2O$, and pressure were interpolated from the National Centers for Environmental Protection (NCEP)
reanalysis data and the *a priori* vertical profiles of other gases were from the statistical averages of
the Whole-Atmosphere Community Climate Model version 6 (WACCM) simulations from 1980 to
2020. The diagonal elements of the *a priori* covariance matrices $\mathbf{S}_a$ and the measurement noise
covariance matrices $\mathbf{S}_\varepsilon$ were set to standard deviation (SD) of the WACCM simulations and the
inverse square of the signal-to-noise ratio (SNR) of each spectrum, respectively (Franco et al. 2015).





The non-diagonal elements of both $\mathbf{S}_a$ and $\mathbf{S}_\varepsilon$ were set to zero. The instrument line shape (ILS) of
the FTIR instrument deduced from optical path alignment diagnosis with a low-pressure HBr cell
was adopted in the retrieval (Hase 2012; Sun et al. 2018b).

4        For each retrieval, the averaging kernels reflect the sensitivity of the retrieved profile to the

real profile. The area of the averaging kernels at a specific height is calculated as the sum of the
elements of the corresponding averaging kernels (Pougatchev et al. 1995). It represents the fraction
of the retrieval at that height which comes from the measurement rather than from the *a priori*
information (Rodgers, 2000). A value close to unity at a specific height indicates that the retrieved
profile at that height is nearly independent of the *a priori* profile and is thus from the measurement.
The trace of the averaging kernel matrix is defined as degrees of freedom for signal (DOFS) and it
quantifies the number of independent information in the retrieved profile. Fig. 1 shows the averaging
kernels as well as their area, cumulative sum of DOFS, and VMR profile of randomly selected $C_2H_6$
retrieval at Hefei. Ground-based FTIR $C_2H_6$ observations at Hefei have a sensitivity of larger than
0.7 from ground to about 10 km altitude, indicating that the retrievals are mainly sensitive to the
troposphere. This also means that the retrieved profile information below 10 km comes for more
than 70% from the measurement, or in other words, that the *a priori* signal impacts the retrieval by
less than 30% [Fig 1(a)]. The typical DOFS obtained at Hefei over the total atmosphere for $C_2H_6$ is
$1.69 \pm 0.29$ ($1\sigma$), meaning that we can roughly provide two pieces of information on the vertical
profile [Fig 1(b)]. The shape of the retrieved profile is heavily weighted toward the lower
troposphere. As shown in Fig.1(c), the $C_2H_6$ concentration decreased by 72.7% with an increase in
the height from surface to 2 km and kept decreasing slowly in the rest part of the atmosphere till
approaching around zero in the stratosphere and above. The $C_2H_6$ partial column below 10 km
accounted for 88.6% of $C_2H_6$ total column. This percentage is expected to show less seasonal
variation since the shape of the retrieved profile is similar to the shape of the *a priori* profile due to
the low DOFS [Fig. 1 (c)]. As a result, in subsequent analysis, the $C_2H_6$ VMRs averaged between
the surface and 10 km are selected as representatives of $C_2H_6$ troDMF. The selected tropospheric
layer (from surface up to 10 km) corresponds to $1.47 \pm 0.2$ ($1\sigma$) of DOFS and can be used with
confidence. This selected layer is totally within the tropopause height (~ 16 km) at Hefei over four
seasons (Sun et al. 2020b). The Hefei site is located in the northeastern margin of a GEOS-Chem
grid cell [Fig. 2 and Table 3]. This selected layer also ensures the line of sights of all observations
are totally within the same grid cell.

32       We calculated the error budget for $C_2H_6$ retrieval at Hefei following the formalism of Rodgers

(2000), and separated all error components into systematic or random errors according to whether
they vary steadily or randomly over consecutive measurements. The random, systematic, and the
combined error budgets for the selected tropospheric layer (from surface up to 10 km) are
summarized in Table 1. The input covariance matrix of temperature is based on the differences
between Sonde and an ensemble of NCEP temperature profiles near Hefei, leading to 2 to 5 K in
the troposphere and 3 to 7 K in the stratosphere. For each interfering gases, the corresponding
covariance matrix is obtained with the WACCM v6 climatology. The input covariance matrix of
measurement error is based on the inverse square of the SNR of each spectrum. We regularly use a
low-pressure HBr cell to diagnose the misalignment of the FTIR spectrometer and to realign the
instrument when indicated. The FTIR spectrometer at Hefei is assumed to be not far from the ideal
condition, and the input uncertainties for zero level, background curvature, field of view, optical
path difference, solar zenith angle, interferogram phase, and ILS are estimated to be 1%. For the


$C_2H_6$ spectroscopic absorption coefficients, the line list in atm16 follows HITRAN 2012 (Rothman
et al., 2013), and we use 5% for line intensity, pressure-, and temperature-broadening coefficients.
For the retrieval parameter and smoothing error, the input covariance matrix are prescribed from the
optimal estimation retrieval outputs. To estimate the retrieval error of $C_2H_6$ troDMF at Hefei, the
elements of all gain matrices were set to zero for the altitudes outside the selected layer. The
contributions of all error components to $C_2H_6$ troDMF retrieval at Hefei are summarized in Table 1.
The dominant random errors are from temperature uncertainty (1.69%) and the zero level
uncertainty (1.54%), and the dominant systematic error is from the line intensity uncertainty
(5.12%). Total random and systematic errors are estimated to be 2.32% and 5.48%, respectively.
Total retrieval error calculated as square root sum of the squares of total random and systematic
errors is estimated to be 6.21%.
In order to exclude the measurements that seriously affected by instable weather conditions or
by the *a priori* profile due to low measurement information content in less favourable observational
conditions, e.g., around noontime when the probed atmosphere is thinner, or in summer when $C_2H_6$
is less abundant. The FTIR measurements saved with a solar intensity variation (SIV) of larger than
10% or retrievals with total DOFS of less than 0.7 or the root-mean-square (RMS) of fitting residuals
of larger than 2%, which accounted for 11.2% of total measurements, were excluded in this study.
**2.2 GEOS-Chem sensitivity simulation**
Relative contribution of various source categories and regions to the observed $C_2H_6$ abundance
were quantified by a series of sensitivity simulations using the GEOS-Chem chemical model version
12.2.1 (Bey et al. 2001) (http://geos-chem.org, last access on 24 August 2020). All simulations
implemented a universal tropospheric-stratospheric Chemistry (UCX) mechanism (Eastham et al.
2014; Fisher et al. 2017) and were driven by the Goddard Earth Observing System-Forward
Processing (GEOS-FP) meteorological fields at a degraded horizontal resolution of 2°×2.5°. The
temporal resolutions are 1 hour (hr) for surface variables and boundary layer height, and 3 hr for
other variables. Dry deposition was calculated by the resistance-in-series algorithm (Wesely 1989;
Zhang et al. 2001) and wet deposition followed that of Liu et al. (2001). The photolysis rates were
available from the FAST-JX v7.0 photolysis scheme (Bian and Prather 2002). All simulations were
spun up for one year (July 2014 to July 2015) and output hourly mean $C_2H_6$ VMR profiles globally
ranging from the surface to 0.01 hPa at a horizontal resolution of 2°×2.5°. This study only
considered the $C_2H_6$ simulations from 2015 to 2020 in the grid box containing Hefei (31.52°–
32.11°N by 116.53°–118.02°E).
In the recent past, the inventories led to significant underestimation of the $C_2H_6$ simulation
(e.g., HTAP2 in Franco et al. 2016). Since then, some efforts have improved the situation (e.g.,
Tzompa-Sosa et al. 2017). In this study, we refer to Sun et al. (2020a) for more details on
implementation of emission inventories. Briefly, global anthropogenic emissions were from the
Community Emissions Data System (CEDS) inventory which overwrites Asia by the latest Multi-
resolution Emission Inventory for China (MEIC) (Hoesly et al. 2018; Li et al. 2017; Lu et al. 2019;
Zheng et al. 2018). Global biomass burning and biogenic emissions were from the Global Fire
Emissions Database (GFED v4) inventory (Giglio et al. 2013) and the Model of Emissions of Gases
and Aerosols from Nature (MEGAN version 2.1) inventory (Guenther et al. 2012), respectively.
The $CH_4$ emission fields are prescribed based on NOAA measurements for 1983–2016 and are



extended to 2020 using the linear extrapolation of local 2011–2016 change rates (Murray, 2016; Lu
et al., 2019).

3       Particular improvements have been done for the latest bottom-up MEIC emission inventory in
the accuracy of vehicle emission modelling (Zheng et al. 2014), power plant emission calculation
(Liu et al. 2015), and the non-methane VOCs (NMVOCs) speciation method (Li et al. 2014). Many
studies have verified that the MEIC emission inventory can reasonably represents the anthropogenic
emissions over Asia (Hoesly et al. 2018; Li et al. 2017; Lu et al. 2019; Sun et al. 2020a; Tian et al.
2018; Yin et al. 2020; Yin et al. 2019; Zheng et al. 2018). Anthropogenic $C_2H_6$ emissions in China
by region and category for the 2015 and 2019 MEIC emission inventories are summarized in Table
2. All subdivided geographical regions are shown in Fig. 2 and the resulting delimitations are
summarised in Table 3. The delimitations of these geographical regions are based on the levels of
urbanization and industrialization in China. Region ① in Fig. 2 only covers a few sparsely city
clusters representing the region with least population and industrialization in China (Lu et al. 2019).
Regions ②, ④, and ⑤ cover the North China Plain (NCP), Yangtze River Delta (YRD), and
Pearl River Delta (PRD) city clusters, respectively, which are the three most developed city clusters
facing severe air pollution in China. Region ③ covers the Sichuan Basin (SCB) and central
Yangtze River (CYR) city clusters with newly emerging severe air pollution in China. Total annual
Chinese anthropogenic emissions of $C_2H_6$ in 2015 and 2019 are 0.883 Tg and 0.859 Tg, respectively.
In both years, anthropogenic $C_2H_6$ emissions in China are dominated by industry and residential
emissions. The highest anthropogenic $C_2H_6$ emission rates (calculated as the ratio of total $C_2H_6$
emission to the coverage) are in densely populated and industrialized region clusters in eastern part
of China (including northern China (NR), eastern China (ER), central China (CR), southern China
(SR), and adjacent regions) with little seasonal variation [Fig.A1]. The anthropogenic emissions in
WR region are typically lower than those in other parts of China because of lower population and
industries in the region (Lu et al. 2019; Zheng et al. 2018).
In order to quantify the contributions of different source categories and regions to the observed
$C_2H_6$, we first conducted a reference full chemistry simulation (BASE) with implementation of all
emission inventories as described above. Then, we conducted a series of sensitivity simulations to
assess the change of each sensitivity simulation relative to the BASE simulation. Model
configurations in this study were similar to those in Sun et al. (2020a) but with a different emission
perturbation method. When an emission inventory was shut off in Sun et al. (2020a), the emissions
of all atmospheric compounds in that inventory were suppressed. In contrast, this work only
suppressed $C_2H_6$ in each case except for biogenic and biomass burning emission perturbations,
which suppressed all atmospheric compounds since we cannot separate $C_2H_6$ emission from current
biogenic and biomass burning emission inventories. Model configurations in this study are
summarised in Table 3 and were described as follows.
(i) To analyse the contributions of different emission categories, we shut off $C_2H_6$ in each
individual emission inventory to evaluate the change of the simulation in the presence of $C_2H_6$ in
other emission inventories. As a result, the relative contribution of each emission category was
estimated as the relative difference between the GEOS-Chem simulation in the presence and
absence of $C_2H_6$ in that emission inventory. We have conducted four such sensitivity simulations by
shutting off $C_2H_6$ emissions in (1) fossil fuel emission inventory (including emissions from
agriculture, industry, power plant, residential, and transport), (2) biogenic emission inventory, (3)
biomass burning emission inventory, and (4) biofuel emission inventory (Table 3). The sum of fossil



fuel and biofuel $C_2H_6$ emissions is defined as anthropogenic $C_2H_6$ source and the sum of biogenic
and biomass burning $C_2H_6$ emissions is referred to as natural $C_2H_6$ source.

3        (ii) To analyse the contributions of different geographical regions, we shut off all categories of
$C_2H_6$ emissions (i.e., the aforementioned anthropogenic plus natural $C_2H_6$ sources) within each
geographical region to assess the change of the simulation in the presence of $C_2H_6$ emissions outside
that geographical region. Thus, the relative contribution of each geographical region was estimated
as the relative difference between the GEOS-Chem simulation in the presence and absence of $C_2H_6$
emissions within that geographical region. We have conducted five such sensitivity simulations by
shutting off $C_2H_6$ emissions within five geographical regions shown in Fig. 2.
**2.3 Generalized additive models (GAMs) regression**

11       In this study, we investigate the dependencies of $C_2H_6$ on meteorological and emission factors
by using the GAMs regression (Wood and Simon 2004; Wood 2004). Regression analysis is
proceeded using the thin plate smoothing spline function (Pearce et al. 2011). Smoothing parameters
and confidence intervals are calculated according to the Restricted Maximum Likelihood standard
(REML) and the unconditional Bayesian method, respectively (Pearce et al. 2011). The GAMs
regression is better than the traditional statistical models in dealing with nonlinear fittings (Veaux
and Richard 2012). For climate change applications, where there are many non-linear relationships
between variables, the GAMs regression is particularly attractive (Zhang et al. 2019).

19       We introduced a variety of potential meteorological and emission factors into the GAMs
regression one at a time and performed significance tests based on the Akaike's Information Criteria
(AIC) values (Wood and Simon 2004). The explanatory variables which passed the significance
tests with the smallest AIC values were included into the final GAMs model. Furthermore,
explanatory variables in GAMs regression may interact with each other and result in unstable
fittings due to the internal multicollinearity. For the explanatory variables that show a strong
collinearity with each other, we only included one of them in the final GAMs model. The degree of
multicollinearity can be quantified by the variance inflation factor (VIF) (Ma et al. 2020). Generally,
a stronger collinearity between the explanatory variables results in a larger VIF, and the VIF of an
explanatory variable could be 1.0 if it is not correlated with other explanatory variables (Ma et al.
2020). In this study, we included all the meteorological and emission factors in the GAMs and
calculated the VIF for all the influencing factors. The multicollinearity diagnosis concluded that the
main causes of multicollinearity are between the HCN and CO, and between the tropopause height
and planetary boundary layer height (PBLH). Including either of these two data pairs in the GAMs
regression showed significant collinearities with the VIF values of greater than an empirical
threshold of 4.0, indicating an unstable regression (Lin et al. 2018). After omitting HCN and PBLH
in the final GAMs model, the adjusted VIF values of all the variables were less than 4.0 and the
variables uniformly passed the significance tests. As a result, the final GAMs model in the context
of the $C_2H_6$ troDMF time series $y$ can be described as (Pearce et al. 2011):

38       $$\log(y) = \beta + S(ua) + S(va) + S(omega) + S(qv) + S(troph) \qquad (1)$$
39       $$+ S(pres) + S(temp) + S(ch_4) + S(co) + \varepsilon$$

where $\beta$ and $\varepsilon$ are the mean response constant and the fitting residual, respectively. $S(ua)$, $S(va)$,
$S(omega)$, $S(qv)$, $S(troph)$, $S(pres)$, $S(temp)$, $S(ch_4)$, and $S(co)$ are the smoothing functions of daily
average zonal wind (with a unit of m s$^{-1}$), meridional wind (m s$^{-1}$), vertical wind (Pa s$^{-1}$), water vapor





concentration (%), tropopause height (km), pressure (hPa), temperature (°C), $CH_4$ troDMF (ppbv),
and CO troDMF (ppbv). Positive values of *ua*, *va*, and *omega* represent northward, eastward, and
upward winds, respectively. The sum of $S(ch_4)$ and $S(co)$ is referred to as the emission influences,
and the sum of remaining smoothing functions is referred to as the meteorological influences.
For driving the GAMs regression, we first derived $CH_4$ and CO VMR profiles from direct solar
absorption spectra similar to that of $C_2H_6$, see section 2.1. The spectra for $CH_4$ retrievals are exactly
the same as those of $C_2H_6$ but the spectra for CO are saved at a different filter channel. The
respective VMR profiles were then converted to troDMF values following the method of $C_2H_6$. The
retrieval configurations, waveband selections and the interfering gases considerations for $CH_4$ and
CO can be found in Sun et al. (2018b). The DOFS of the retrievals between surface and 10 km
altitude for both $CH_4$ and CO are larger than 1.5 and the corresponding retrieval errors are less than
8% (Sun et al., 2018b). All meteorological factors are from the GEOS-FP meteorological fields at
their native resolution of 0.25° × 0.3125° ranging from the surface to 0.01 hPa at a temporal
resolution of 1 hr. Since the meteorological fields and $C_2H_6$ concentration are not uniformly
distributed along the altitude, the summing mean values of the meteorological fields cannot properly
characterize the meteorological influences. In this study, we use a method similar to that of
Shaiganfar et al. (2017) to increase the influence weighting toward lower troposphere. As a result,
all meteorological parameters except tropopause height (*troph*) are converted into the $C_2H_6$ profile
weighting averaged value $\omega_{avg}$ through Eq. (2):

$$\omega_{avg} = \frac{\sum_i \boldsymbol{\omega}(z_i) \cdot \boldsymbol{c}(z_i) \cdot \boldsymbol{Airmass}(z_i)}{\sum_i \boldsymbol{c}(z_i) \cdot \boldsymbol{Airmass}(z_i)} \quad (2)$$

where $\boldsymbol{\omega}(z_i)$, $\boldsymbol{c}(z_i)$, and $\boldsymbol{Airmass}(z_i)$ represent the value of the meteorological factor, $C_2H_6$
concentration, and the air mass at the altitude $z_i$.
**3. Variability and comparison with GEOS-Chem model data**
We have compared the observed daily mean time series and seasonal cycle of $C_2H_6$ troDMF to
the GEOS-Chem BASE simulations for investigating the chemical model performance for the
specifics of polluted regions over eastern China. As the vertical resolution of GEOS-Chem is
different from the FTIR observation, smoothing correction has been done for the GEOS-Chem
profiles (Rodgers and Connor 2003). First, the GEOS-Chem daily mean profiles of $C_2H_6$ have been
interpolated to the FTIR altitude grid for ensuring a common altitude grid. In order to match the
FTIR observations which only operates during daytime, the average for GEOS-Chem simulations
was only performed during daytime from 9:00 to 17:00 local time (LT). The interpolated profiles
were then smoothed by the concurrent seasonal mean values of the FTIR averaging kernels and *a*
*priori* profiles (Rodgers 2000; Rodgers and Connor 2003). The smoothed GEOS-Chem profiles
were subsequently converted to troDMF values by using the corresponding regridded air density
profiles from the model with the method described in section 2.1. Finally, the GEOS-Chem $C_2H_6$
troDMF time series only for the days with available FTIR observations were averaged by month
and compared with the FTIR monthly mean data.
Fig. 3 (a) shows the comparison of daily mean time series of $C_2H_6$ troDMF between the FTIR
observation and the smoothed GEOS-Chem model simulation from 2015 to 2020. Fig. 3 (b)
compares the seasonal cycles derived from Fig. 3 (a) for the days with available FTIR observations
only. Generally, the measured features in terms of seasonality and interannual variability can be





reproduced by the GEOS-Chem simulations with a correlation coefficient ($r$) of 0.88. Large GEOS-
Chem vs. FTIR differences tended to occur in the trough and peak of the observations. For instance,
the observed monthly mean value of $C_2H_6$ troDMF was overestimated by 35.6% in July and
overestimated by 17.4 % in December by the GEOS-Chem. These discrepancies may be mainly
attributed to uncertainties in the horizontal transport and vertical mixing schemes simulated by the
GEOS-Chem model at a relatively coarse spatial resolution, which are difficult to match column
observation over a single point. In addition, the number of $C_2H_6$ measurements at Hefei on each
measurement day varied a wide range from 1 to 17 depending on the weather condition, but GEOS-
Chem simulations are available consecutively by hour for the same day. This difference in the
temporal resolution of GEOS-Chem and FTIR could also cause inconsistencies due to the high
variability of atmospheric $C_2H_6$. However, considering the concurrent data pairs only (± 30 min),
the averaged difference between GEOS-Chem and FTIR data (GEOS-Chem minus FTIR) is (−0.02
± 0.05) ppbv (−1.6 ± 4.2) %, which is within the FTIR uncertainty budget. As a result, GEOS-Chem
can simulate $C_2H_6$ concentration and variability for the specifics of polluted regions over eastern
China. With improved emission inventories, previous studies have also found that global chemistry
transport models were able to reproduce the absolute values as well as seasonal cycles of the ground-
based FTIR $C_2H_6$ observations in the other parts of the world (Franco et al., 2015; Franco et al.,
2016; Tzompa-Sosa et al. 2017).
As typically observed, the peak-to-peak amplitude of the modulation with respect to the
monthly mean of $C_2H_6$ troDMF spanned a large range of −16.0% to 72.8% depending on season
[Fig. 3 (b)]. The observed $C_2H_6$ troDMF roughly decreases over time for the first half of the year
and increases over time for the second half of the year [Fig. 3 (b)]. High levels of $C_2H_6$ troDMF
occur in the late autumn to early spring and low levels of $C_2H_6$ troDMF occur in late spring to early
autumn. The observed $C_2H_6$ troDMF reached a minimum monthly mean value of (0.36 ± 0.26) ppbv
in July and a maximum monthly mean value of (1.76 ± 0.35) ppbv in December. $C_2H_6$ troDMFs in
December were on average 4.9 times higher than those in July. Since the tropospheric OH oxidation
capability in summer is higher than that in winter, the $C_2H_6$ seasonality characterized by a winter
maximum and a summer minimum was also observed in other FTIR stations (Angelbratt et al. 2011;
Franco et al. 2015; Franco et al. 2016; Lutsch et al. 2019; Nagahama and Suzuki 2007; Rinsland et
al. 2002; Simpson et al. 2012; Viatte et al. 2015; Viatte et al. 2014; Vigouroux et al. 2012; Zeng et
al. 2012; Zhao et al. 2002). We have used the bootstrap resampling method of Gardiner et al. (2008)
to evaluate the seasonality and interannual variability of $C_2H_6$ troDMF, where a 3$^{rd}$ Fourier series
plus a linear function was used to fit daily mean time series of $C_2H_6$ troDMF by both FTIR
observations and GEOS-Chem model simulations. We incorporated the errors arising from the
autocorrelation in the residuals into the uncertainties in the change rates following the procedure of
Santer et al. (2008). The observed $C_2H_6$ troDMFs from 2015 to 2020 showed a negative change rate
of (−2.60 ± 1.34)% yr$^{-1}$, which is in reasonable agreement with the modelled change rate of (−2.1 ±
0.7)% yr$^{-1}$ [Fig. 3 (a)].
**4. GAMs regression results and interpretation**
$C_2H_6$ troDMF time series from 2015 to 2020 over Hefei by the FTIR and the GAMs regression
model are shown in Fig. 4. The observed $C_2H_6$ variability can be reproduced by the GAMs
regression model with a good agreement, as confirmed by a correlation coefficient ($r$) of 0.90 [Fig.





4(a)]. Meanwhile, the observed $C_2H_6$ troDMF time series also show high correlations with both the
accumulated meteorological factor ($r$=0.88) [Fig. 4 (b)] and the accumulated emission factor ($r$ =
0.70) [Fig. 4(c)], indicating both meteorological and emission influences are important factors for
driving the $C_2H_6$ troDMF variability. Generally, both meteorological and emission factors show
positive influences on $C_2H_6$ troDMFs in cold season (DJF/MAM) and negative influences on $C_2H_6$
troDMFs in warm season (JJA/SON). However, the seasonal dependency of meteorological
influence is stronger than that of emission influence. During the studied years, the year to year
differences in meteorological influence are small, while the positive emission influence showed an
overall decreasing change rate since 2016, which probably drives a decreasing change rate in $C_2H_6$
troDMF in recent years.
The influence of each explanatory variable $x_i$ in GAMs regression calculated as $100\% \cdot [e^{s(x_i)}-1]$
are shown in Fig. 5, which reflects the influence of each individual variable on the relative change
of $C_2H_6$ troDMF. The corresponding DOFS of each smoothing function are also shown in Fig.5. If
an explanatory variable is linearly correlated with the $C_2H_6$ troDMF, the DOFS of the resulting
smoothed variable could be equal to 1, and the larger the slope the higher the linear response.
Otherwise, the larger the deviation of DOFS relative to 1, the more significant the nonlinear
relationship is (Veaux and Richard 2012). During the studied years, the DOFS of zonal wind (*ua*),
convection wind (*omega*), pressure (*pres*), tropopause height (*troph*), temperature (*temp*), and CO
troDMF (*co*) were close to 1, reflecting a roughly linear relationship of these explanatory variables
with the $C_2H_6$ troDMF. In contrast, the DOFS of meridional wind (*va*), $H_2O$ troDMF (*qv*) and $CH_4$
troDMF (*ch4*) were much greater than 1, reflecting a significant nonlinear relationship with the $C_2H_6$
troDMF.
The observed $C_2H_6$ troDMF was influenced by many factors. The zonal wind (*ua*), meridional
wind (*va*), $CH_4$ troDMF (*ch4*), and CO troDMF (*co*) showed positive influences and the other
explanatory variables showed negative influences on the observed $C_2H_6$ troDMF variability. The
results show that the northward, eastward, and downward convection winds facilitate the
accumulation of $C_2H_6$ over the observation site and result in higher $C_2H_6$ troDMF. Since most
anthropogenic and biomass burning sources of CO, and fossil fuel source of $CH_4$ are common
sources of $C_2H_6$, $C_2H_6$ troDMF gradually went up as $CH_4$ and CO troDMFs increased. In contrast,
meteorological conditions as high temperature, high humidity, and low pressure are more favorable
to $C_2H_6$ oxidation, resulting in lower $C_2H_6$ troDMF. Meanwhile, deep upward convection wind and
high tropopause height facilitate the diffusion of $C_2H_6$ over the observation site and result in lower
$C_2H_6$ troDMF.
**5. Source attribution**
**5.1 Contributions of different source categories and regions**
The absolute and relative seasonal contributions of fossil fuel, biogenic, biomass burning, and
biofuel emissions to $C_2H_6$ abundance from 2015 to 2020 over Hefei are shown in Fig. 6. The GEOS-
Chem annual mean $C_2H_6$ troDMF simulations were decreased by 0.53, 0.26, 0.31, and 0.20 ppbv in
the absence of fossil fuel, biogenic, biomass burning, and biofuel $C_2H_6$ emission inventories, which
correspond to 34.5, 14.7, 22.4, and 14.7% of relative contribution to the modelled $C_2H_6$ abundance,
respectively. The anthropogenic emissions account for 49.2% and the natural emissions account for
37.1% of the $C_2H_6$ abundance. The remaining contribution calculated as the difference between the



BASE simulation and the sum of anthropogenic and natural contributions is 0.17 ppbv (13.7%).
This missing rest can be largely attributed to nonlinear interactional effects among different sources
which were not captured by the sensitivity simulations. Indeed, shutting off an emission inventory
may induce significantly lower concentrations in atmospheric compounds (i.e., $C_2H_6$ for noFF and
noBIOF or all suppressed compounds for noBVOC and noBB simulations) globally. On the one
hand, some of them may react with OH, which would lead to higher OH concentrations available
for the oxidation of $C_2H_6$, and eventually enhances the $C_2H_6$ destruction from other emission
categories. On the other hand, some of them may form OH by a series of oxidation reaction, which
would lead to lower OH concentrations available for the oxidation of $C_2H_6$, and eventually mitigates
the $C_2H_6$ destruction from other emission categories. However, it is difficult to quantify the
nonlinear impact of each individual emission category, since the concentration and spatial
distribution of $C_2H_6$ in each emission category are different. Especially when biogenic and biomass
burning emissions are suppressed, the impacts become hard to assess, since all NMVOCs
compounds play a key role in both OH formation and destruction. Investigating the nonlinear impact
of each individual emission category would require additional work that is beyond the scope of the
present work.

17        The contributions of all emission sources are seasonal dependent, and the fossil fuel

contribution shows the largest seasonal difference, which consolidates the GAMs regression results
that the emission influences are seasonal dependent. The fossil fuel contribution in winter and spring
(DJF/MAM) are larger than those in summer and autumn (JJA/SON), with a maximum of 52.0% in
DJF and a minimum of 13.0% in JJA. The JJA/SON meteorological conditions which show stronger
solar radiation, higher temperature, wetter atmospheric condition, and lower pressure than those in
DJF/MAM are more favorable for increasing VOCs emissions from biogenic sources (BVOCs),
which consolidates the fact that $C_2H_6$ abundance from biogenic source in JJA/SON are larger than
those in DJF/MAM. The missing rest contributes to a maximum of 32% in JJA when the $C_2H_6$
oxidation reaches the seasonal maximum and is thus more sensitive to the on-off state of different
sources.

28        Fig. 7 explores the absolute and relative seasonal contributions of ER, CR, NR, WR, and SR

regions to the $C_2H_6$ abundance from 2015 to 2020 over Hefei. The GEOS-Chem annual mean $C_2H_6$
troDMF simulations were decreased by 0.28, 0.22, 0.29, 0.07, and 0.12 in the absence of the $C_2H_6$
emissions in ER, CR, NR, WR, and SR regions, which correspond to 21.5%, 15.8%, 20.3%, 5.7%,
and 8.9%, of relative contribution to the modelled $C_2H_6$ abundance, respectively. The contributions
of all geographical regions are also seasonal dependent. The results show that the observed $C_2H_6$
abundance was largely attributed to emissions within China (74.1%), which show a maximum in
JJA/SON and a minimum in DJF/MAM. As vicinities of the observation site, the ER, CR, and NR
regions dominated the contribution within China (57.6%). The remaining contribution calculated as
the difference between the BASE simulation and the total contributions of above individual source
regions is 0.42 ppbv (25.9%). This contribution is the sum of $C_2H_6$ emissions outside China (ROW)
and the nonlinear interactional effects among the geographical sensitivity simulations. This rest
contribution in DJF/MAM are ~ 4.0 times larger than those in JJA/SON. Considering the nonlinear
interactional effects mainly occur in JJA/SON but this rest contribution in the meantime shows the
seasonal minimum value, this remaining contribution can be largely attributed to ROW
contributions.

44        As a relatively long lifetime species (a few months), $C_2H_6$ emissions originating from either


nearby or in distant areas can be transported to Hefei site under favourable weather conditions, and
thus contribute to the observed $C_2H_6$ abundance. In addition, atmospheric compounds originating
either nearby or in distant areas, which affect the chemistry of $C_2H_6$ oxidation, could also affect the
observed $C_2H_6$ abundance. For contributions within China, the lowest contribution of the WR region
to the observed $C_2H_6$ abundance is largely attributed to the lowest $C_2H_6$ emission rates in this region
(Table 2). A smaller contribution of the SR region to the observed $C_2H_6$ abundance in DJF/MAM in
comparison with the ER, CR, and NR regions is mainly attributed to less air masses that originated
in south China under the dry winter monsoon conditions, see section 5.2.
**5.2 Transport inflow and outflow pathways**
The direct GEOS-Chem sensitivity simulations can clearly characterize the 3D transport inflow
and outflow pathways of $C_2H_6$ over the observation site. Fig.8 shows the spatial distribution of
GEOS-Chem $C_2H_6$ BASE simulations around China along with horizontal wind vectors at 900 hPa
in different seasons. General atmospheric circulation patterns over eastern China are typically
affected by mid-latitude westerlies and Asian monsoon, including the East Asian summer monsoon
and South Asian summer monsoon (Chen et al. 2009; Liang et al. 2005; Liu et al. 2003). Fig. 9
illustrates the latitude − height distributions of $C_2H_6$ VMR over China from six source regions along
with the 3D atmospheric circulation patterns in different seasons. The 3D transport inflow and
outflow pathways of $C_2H_6$ over the observation site are thus deduced as follows.
As indicated by the arrows in Figs. 8 and 9, the high pressure system over the Eurasian
continent in DJF triggers the descending of strongly cold air over eastern China and results in air
masses converging toward the observation site from western and northern areas, while the high
pressure system over the Indian ocean and Pacific in JJA triggers the ascending of strongly heated
air over eastern China and results in air masses converging toward the observation site from South
Asia and East Asia (SEAS) (Liang et al. 2004; Liu et al. 2003). In DJF, the mid-latitude westerlies
extend to tropics (about 15 °N) over middle and upper troposphere, and subtropics (about 30°N)
near the surface, while the easterlies mainly prevail in tropics and are weak over eastern China [Figs.
8 and 9]. In the summer monsoon season, the atmospheric circulation patterns over eastern China
change dramatically and is dominated by surface wind regime originating from Pacific, South China
Sea, or Arabian Sea [Figs.8 and 9]. Meanwhile, the mid-latitude westerlies recede to the North
Temperate Zone (north of 30°N) and the westerly jet center shifts to north of 50°N in JJA (from ~
30°N in DJF) [Fig. 8]. In JJA, the tropical region is characterized by the strong easterlies in the
upper troposphere and by the southwesterly air flow in the lower troposphere [Fig. 9]. The prevailing
winds in the transition seasons in MAM and SON are still westerlies with frequent cold fronts [Figs.
8 and 9]. These above seasonal circulation patterns determine the transport inflow and outflow of
$C_2H_6$ around the observation site. However, the transported scales are also influenced by source
location and strength, travel trajectory and travel time (Liu et al. 2003).
Generally, $C_2H_6$ emissions from CR, NR, and WR regions can be transported to the observation
site by the strong westerlies throughout the year [Fig. 9]. $C_2H_6$ from the SR region can be transported
to the observation site by deep convection followed by northward transport into the mid-latitude
westerlies in MAM or driven by the South Asian summer monsoon or westerlies in other seasons
(Liu et al. 2003)[Fig. 8]. The observed $C_2H_6$ transport inflow originating from the local ER region
is mainly driven by the local circulation cell or Asian monsoon. The driver for $C_2H_6$ transport inflow





originating from the northern ROW (> 32°N) is the same as those from the CR, NR, and WR regions.
The driver for $C_2H_6$ transport inflow originating from the southern ROW (< 32°N) is the same as
that from the SR region. However, $C_2H_6$ originating from ROW relative to those from China reaches
a higher altitude due to longer transport distances. The ROW contributions in JJA/SON are lower
than those in DJF/MAM can be partly attributed to a stronger removal along the long-range transport
pathway by the abundant wet precipitation and oxidation during the summer monsoon and post
monsoon season.
Seasonal variability in $C_2H_6$ transport outflow over the observation site is mainly associated
with the monsoon system. In DJF, $C_2H_6$ over the observation site is capped in the lower troposphere
by the subsidence over eastern China and is swept by northeasterly southwestward into southwestern
areas, where they are lifted up into the free troposphere by convection and then flow away
northeastward [Fig. 9]. In JJA, the observed $C_2H_6$ is transported northeastward by the Asian
monsoon and is undergone frequent lifting into the upper troposphere by deep convection [Figs. 8
and 9]. Frequent cold fronts are common phenomena during meteorologically transitional periods
in MAM and SON. In SON, the winter monsoon builds continental high system, and sweeps the
observed $C_2H_6$ in the lower troposphere northward to relatively high latitudes where they can be
lifted up into the free troposphere by deep convection or cold fronts. In MAM, convection over
lower latitudes at Asian continent starts to rise, which lifts up the observed $C_2H_6$ into the free
troposphere and then flow away southward.

## 5.3 Potential factors driving interannual variability of $C_2H_6$

China has implemented a series of active clean air policies since 2013 to mitigate severe air
pollution problems ( Sun et al. 2020b; Zhang et al. 2019; Zheng et al. 2018). Since then the emissions
of major air pollutants have decreased, and the overall air quality has greatly improved (Sun et al.
2020b; Zhang et al. 2019; Zheng et al. 2018). Many air pollutants, such as $NO_2$, sulphur dioxide
($SO_2$), particulate matter 2.5 ($PM_{2.5}$), $PM_{10}$, CO, black carbon (BC), and organic carbon (OC),
showed negative trends in recent years (Lu et al., 2019; Zhang et al. 2019; Zheng et al. 2018). We
follow the method of Zheng et al. (2018) to estimate the relative change rate of anthropogenic $C_2H_6$
emissions in China during 2015 – 2019 using the MEIC emission inventory. As tabulated in Table
2, anthropogenic $C_2H_6$ emissions in all geographical regions showed a decreasing change rate during
2015 – 2019 except those in WR region where industrialization, urbanization, land use, and
infrastructure construction have expanded rapidly in recent years, resulting in an increasing change
rate of anthropogenic $C_2H_6$ emissions in the region (Ran et al. 2014). The relative change rates of
anthropogenic $C_2H_6$ emissions in China during 2015 – 2019 are estimated as: 12.12% for WR, –
5.32% for NR, –1.03% for CR, –7.66% for ER, –5.01% for SR, and –2.74% in total. The major
driving forces for the decline in $C_2H_6$ emissions over China are attributed to the reductions from
residential and transport sectors, with relative change rates of –14.53% and –3.53%, respectively
(Table 2).
The interannual change rate of anthropogenic $C_2H_6$ emissions in China in recent years is
estimated to be – 0.69% yr$^{-1}$ (calculated as – 2.74%/(2019 – 2015)), which is much lower than the
observed decreasing change rate in $C_2H_6$ tropDMF ($-2.60 \pm 1.34\%$ yr$^{-1}$), indicating additional
driving forces could exist, e.g., reductions in natural $C_2H_6$ emissions in China or reductions in long
range transport of $C_2H_6$ emissions originating either anthropogenic or natural sources outside China.





On the one hand, the Law of the People's Republic of China on the Prevention and Control of Atmospheric Pollution included the prohibition of crop residue burning over China in 2015 because crop residue burning emissions can result in poor air quality (http://www.chinalaw.gov.cn, last access on 19 June 2020). Since then the crop residue burning events over China decreased dramatically (Sun et al. 2020b). Meanwhile, biomass burning events in Africa, SEAS, and Oceania regions in 2015 were higher than those in onward years due to the El Niño Southern Oscillation (ENSO) (Sun et al. 2020b). The decreased global biomass burning emissions could probably also contribute to the observed decreasing change rate in $C_2H_6$ tropDMF over Hefei since 2015. On the other hand, large oil price fluctuations in recent years probably tightened oil and gas development around the globe which can cause a reduction in $C_2H_6$ leakage around the globe. However, $C_2H_6$ observations around the globe and more statistical data are needed to support this deduction, which is beyond the scope of this paper and requires further study.

## 6. Summary and conclusion

Ethane ($C_2H_6$) is an important greenhouse (GHG) gas and plays a significant role in tropospheric chemistry and climate change. As a relatively long residence time species (a few months), observations of $C_2H_6$ can reflect regional and hemispheric changes in emissions and climate, and can be assimilated into a chemical transport model to assess nonlocal emissions and provide valuable insights into model biases of $C_2H_6$ simulations.

This study for the first time presents and quantifies the variability, source, and transport of $C_2H_6$ over densely populated and industrialized eastern China by using ground-based high-resolution Fourier transform infrared (FTIR) observations. Seasonal and interannual variability of $C_2H_6$ over Hefei, eastern China from 2015 – 2020 have been investigated. The dependencies of $C_2H_6$ on meteorological and emission factors were analyzed by using generalized additive models (GAMs). The FTIR $C_2H_6$ time series are for the first time applied to evaluate the standard GEOS-Chem full-chemistry model for the specifics of $C_2H_6$ simulation over eastern China. GEOS-Chem model simulation with the state-of-the art inventory is in good agreement with the FTIR observation. The GEOS-Chem model was further run in a sensitivity mode to quantify relative contribution of various source categories and regions to the observed $C_2H_6$ abundance. The three-dimensional (3D) transport inflow and outflow pathways of $C_2H_6$ over the observation site were finally determined by the GEOS-Chem sensitivity simulation and atmospheric circulation pattern.

We obtained a retrieval error of $6.21 \pm 1.2$ ($1\sigma$)% and degrees of freedom (DOFS) of $1.47 \pm 0.2$ ($1\sigma$) in retrieval of $C_2H_6$ tropospheric column-averaged dry-air mole fraction (troDMF). The observed $C_2H_6$ troDMF reached a minimum monthly mean value of $(0.36 \pm 0.26)$ ppbv in July and a maximum monthly mean value of $(1.76 \pm 0.35)$ ppbv in December, and showed a negative change rate of $(-2.60 \pm 1.34)$ %/yr from 2015 to 2020. Generally, both meteorological and emission factors have positive influences on $C_2H_6$ troDMF in cold season (DJF/MAM) and negative influences in warm season (JJA/SON). GEOS-Chem model sensitivity simulations concluded that the anthropogenic emissions (fossil fuel plus biofuel) accounted for 49.2% and the natural emissions (biomass burning plus biogenic) accounted for 37.1% of $C_2H_6$ abundance over Hefei. The observed $C_2H_6$ abundance over Hefei was mainly attributed to the emissions within China (74.1%), where central, eastern, and northern China dominated the contribution (57.6%). Seasonal variability in $C_2H_6$ transport inflow and outflow over the observation site is largely related to the mid-latitude



westerlies and Asian monsoon system. Reduction in $C_2H_6$ from 2015 to 2020 mainly results from
the decrease in local and transported $C_2H_6$ emissions, which points to air quality improvement in
China in recent years.
4         This study can not only enhance the insights of regional emission, transport, and air clean
actions over China, but also contribute to form new reliable remote sensing dataset in this sparsely-
monitored regions for climate change research.
***Code and data availability.*** The new ground-based Fourier transform infrared (FTIR) spectroscopic
remote sensing dataset for atmospheric $C_2H_6$ over Hefei, eastern China in this study can be accessed
from https://doi.org/10.6084/m9.figshare.13020545. The MEIC emission inventories used in this
study are available from http://meicmodel.org/.
***Author contributions.*** YS designed and wrote the paper with inputs from all coauthors. HY carried
out the GEOS-Chem simulations and GAMs regression. BZ provided the latest MEIC emission
inventory. The rest authors contributed to this work by providing constructive comments.
***Competing interests.*** The authors declare that they have no conflict of interest.
***Acknowledgements.*** The processing and post processing environment for SFIT4 are provided by
National Center for Atmospheric Research (NCAR), Boulder, Colorado, USA. The NDACC
network is acknowledged for supplying the SFIT software. The LINEFIT code is provided by Frank
Hase, Karlsruhe Institute of Technology (KIT), Institute for Meteorology and Climate Research
(IMK-ASF), Germany. We thank the senate of Bremen, Germany for support. We thank the FTIR
group at university of Wollongong, Australia for help in setting up and operating the FTIR
spectrometer at Hefei. We thank the GEOS-Chem team and Tsinghua University, China for
providing the latest MEIC inventory.
***Financial support.*** This work is jointly supported by the National Key Research and Development
Program of China (No.2019YFC0214802, No.2017YFC0210002, No. 2016YFC0203302,
2018YFC0213201, 2019YFC0214702, 2016YFC0200404), the National Science Foundation of
China (No.41775025, No. 41575021, No. 51778596, No. 91544212, No. 41722501, No. 51778596),
and the Sino-German Mobility programme (M-0036). Emmanuel Mahieu is a Senior Research
Associate with the Fonds de la Recherche Scientifique -FNRS. His contribution has been primarily
supported by the F.R.S.-FNRS under Grant no J.0147.18.

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



**Tables**
**Table 1.** Error budget and degrees of freedom (DOFS) for signal of randomly selected $C_2H_6$ troDMF retrieval at
Hefei, eastern China

| Error source | Input value | Error budget |
|---|---|---|
| Temperature uncertainty | $1\sigma$ of NCEP | 1.69% |
| Zero level uncertainty | 1% | 1.45% |
| Retrieval parameters uncertainty | * | < 0.1% |
| Measurement error | $1/SNR^2$ | 0.53% |
| Interfering species uncertainty | SD of WACCM | 0.11% |
| Smoothing uncertainty | * | 0.37% |
| **Total random error** | / | 2.32% |
| Background curvature uncertainty | 1% | 0.14% |
| Field of view uncertainty | 1% | < 0.1% |
| Optical path difference uncertainty | 1% | < 0.1% |
| Solar zenith angle uncertainty | 1% | < 0.1% |
| Phase uncertainty | 1% | < 0.1% |
| ILS uncertainty | 1% | < 0.1% |
| Line temperature broadening uncertainty | 5% | 0.4% |
| Line intensity uncertainty | 5% | 5.12% |
| Line pressure broadening uncertainty | 5% | 0.93% |
| **Total systematic error** | / | 5.48% |
| **Total errors** | / | 6.21% |
| **DOFS (-)** | / | 1.47 |

* These input values for error budget estimation are based on the retrieval output
**Table 2.** Anthropogenic $C_2H_6$ emissions in China by region and category for the 2015 and 2019 MEIC emission
inventories

| Region | | Industry (Tg yr$^{-1}$) | Power plant (Tg yr$^{-1}$) | Residential (Tg yr$^{-1}$) | Transport (Tg yr$^{-1}$) | Sum (Tg yr$^{-1}$) |
|---|---|---|---|---|---|---|
| WR | 2015 | 0.084 | <0.01 | 0.011 | <0.01 | 0.1 |
| | 2019 | 0.097 | <0.01 | 0.011 | <0.01 | 0.112 |
| | change | 15.36% | 82.54% | − 6.61% | − 3.41% | 12.12% |
| NR | 2015 | 0.241 | <0.01 | 0.125 | 0.026 | 0.394 |
| | 2019 | 0.241 | <0.01 | 0.105 | 0.025 | 0.373 |
| | change | 0.04% | 2.51% | − 15.96% | − 4.38% | − 5.32% |
| CR | 2015 | 0.144 | <0.01 | 0.041 | <0.01 | 0.189 |
| | 2019 | 0.15 | <0.01 | 0.033 | <0.01 | 0.187 |
| | change | 4.68% | − 7.00% | −20.75% | − 1.13% | − 1.03% |
| ER | 2015 | 0.07 | <0.01 | 0.026 | 0.01 | 0.11 |
| | 2019 | 0.067 | <0.01 | 0.022 | 0.01 | 0.097 |
| | change | −4.83% | 5.40% | −16.79% | − 3.70% | −7.66% |
| SR | 2015 | 0.06 | <0.01 | 0.026 | 0.01 | 0.09 |
| | 2019 | 0.056 | <0.01 | 0.027 | 0.01 | 0.09 |
| | change | − 7.94% | − 9.26% | 1.52% | − 4.24% | − 5.01% |
| China | 2015 | 0.60 | <0.01 | 0.231 | 0.05 | 0.883 |
| | 2019 | 0.612 | <0.01 | 0.197 | 0.048 | 0.859 |
| | change | 1.91% | 4.04% | −14.53% | −3.93% | −2.74% |


**Table 3.** GEOS-Chem model configurations and delimitations of all geographical regions used in sensitivity
simulations.

| Simulation | Region | Description |
|---|---|---|
| BASE | Global | Standard simulation with all anthropogenic and natural $C_2H_6$ emissions. The BASE simulation is taken as the reference and used for model evaluation |
| noFF | Global | Turn off global fossil fuel $C_2H_6$ emissions in BASE simulation |
| noBVOC | Global | Turn off global biogenic $C_2H_6$ emissions in BASE simulation |
| noBB | Global | Turn off global biomass burning $C_2H_6$ emissions in BASE simulation |
| noBIOF | Global | Turn off global biofuel $C_2H_6$ emissions in BASE simulation |
| Rest | Global | Difference between BASE and the sum of FF, BVOC, BB, and BIOF contributions |
| noWR | 78.6° E – 103.4° E; 27.6°N - 48.8°N | Turn off fossil fuel, biogenic, biomass burning, and biofuel $C_2H_6$ emissions within western China (WR), i.e., region ① in Fig. 2, in BASE simulation |
| noNR | 103.4°E – 129.8°E; 34.6°N – 53.5°N | Turn off fossil fuel, biogenic, biomass burning, and biofuel $C_2H_6$ emissions within northern China (NR), i.e., region ② in Fig. 2, in BASE simulation |
| noCR | 103.4°E – 115.6°E; 27.6°N – 34.6°N | Turn off fossil fuel, biogenic, biomass burning, and biofuel $C_2H_6$ emissions within central China (CR), i.e., region ③ in Fig. 2, in BASE simulation |
| noER | 115.6°E – 123.6°E; 21.0°N – 34.6°N | Turn off fossil fuel, biogenic, biomass burning, and biofuel $C_2H_6$ emissions within eastern China (ER), i.e., region ④ in Fig. 2, in BASE simulation |
| noSR | 98.1°E – 115.6°E; 21.0°N – 27.6°N | Turn off fossil fuel, biogenic, biomass burning, and biofuel $C_2H_6$ emissions within southern China (SR), i.e., region ⑤ in Fig. 2, in BASE simulation |
| ROW | Rest of world | Difference between BASE and the sum of WR, NR, CR, ER, and SR contributions |

# 1 Figures

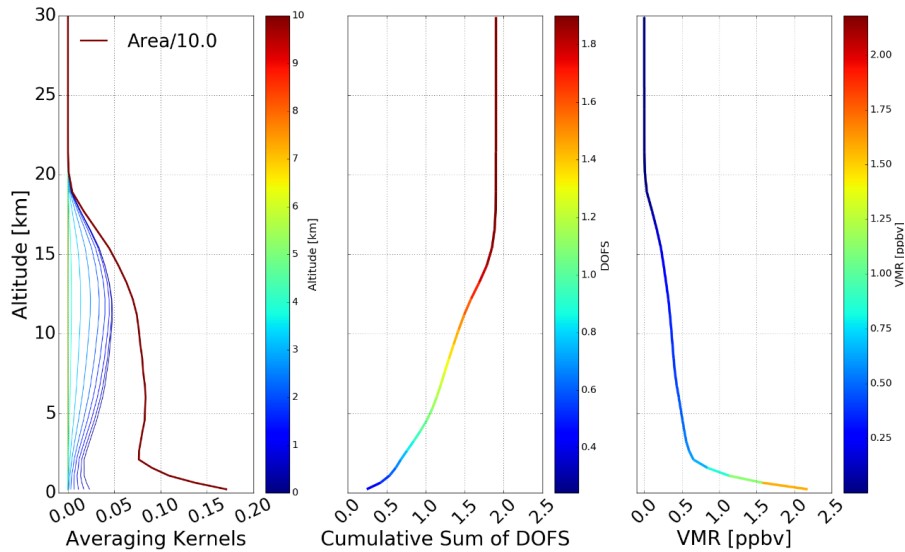

**Fig. 1.** Averaging kernels and their area scaled by a factor of 0.1, cumulative sum of degrees of freedom for signal
(DOFS), and volume mixing ratio (VMR) profile of randomly selected $C_2H_6$ retrieval over Hefei, eastern China.

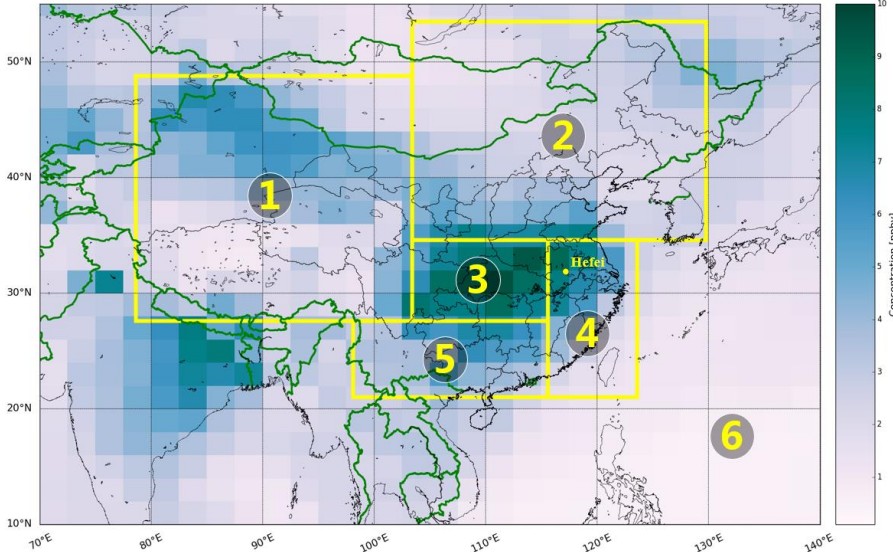

**Fig. 2.** Geographical regions used for GEOS-Chem sensitivity simulations. The numbers ①−⑥ represent western,
northern, central, eastern, and southern China, and the rest of world, respectively. See Table 3 for latitude and
longitude delimitations. Daily mean values of $C_2H_6$ troDMF on 1 January 2017 provided by GEOS-Chem BASE
simulation was selected as a representative of wintertime enhancement in eastern China.

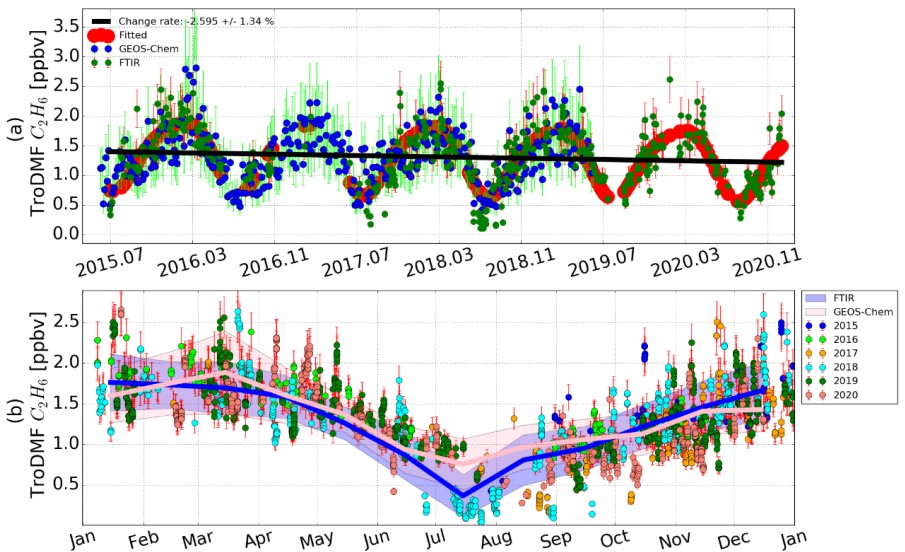

**Fig. 3.** (a) C₂H₆ troDMF time series comparison between FTIR observation and GEOS-Chem model BASE simulation from 2015 to 2020 over Hefei, eastern China. The seasonality and interannual variability are represented by red dots and black line, respectively, which are fitted by using a bootstrap resampling model with a 3$^{rd}$ Fourier series plus a linear function. (b) Seasonal variations of C₂H₆ troDMF by FTIR and GEOS-Chem simulation. Bold curves and the shadows are monthly mean values and the 1-σ standard variations, respectively. Vertical error bars for FTIR and GEOS-Chem represent retrieval uncertainties and diurnal variabilities, respectively.

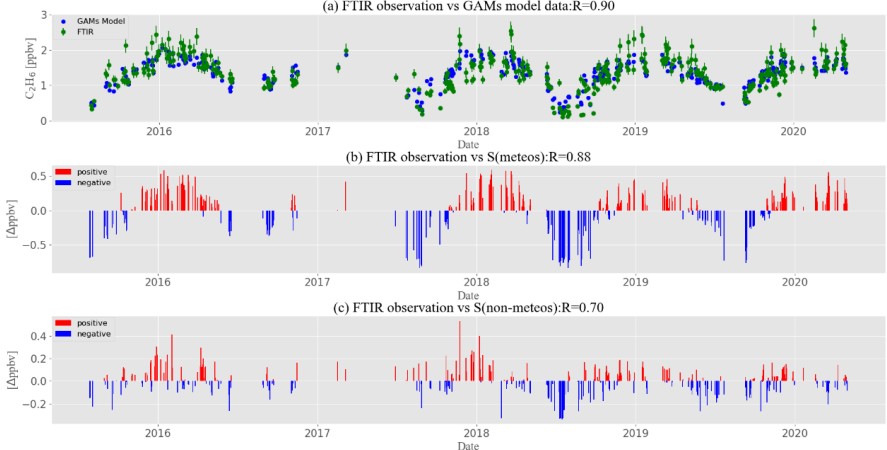

**Fig. 4.** (a) C₂H₆ troDMF time series from 2015 to 2020 over Hefei, eastern China by FTIR and GAMs regression model. (b) Time series of accumulated meteorological smooth functions (S(*meteos*)), and (c) time series of accumulated emission smooth functions (S(*non-meteos*)). Positive and negative influences are indicated with red and blue bars, respectively. Correlation coefficients for the total, meteorological, and emission influences are also shown.

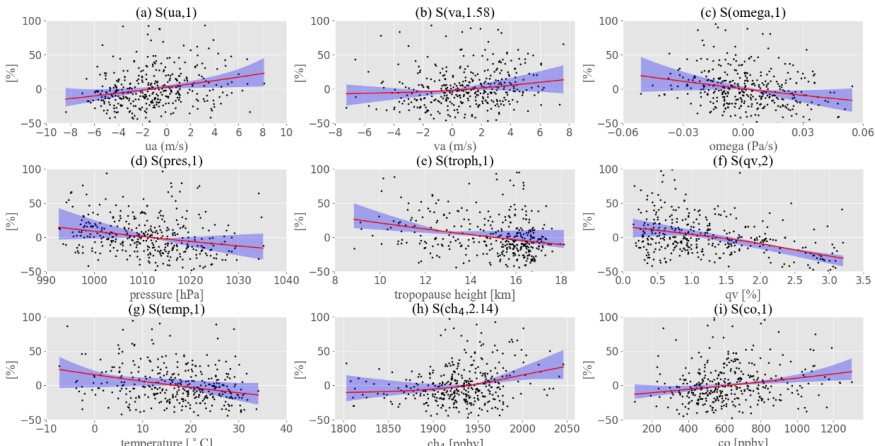

**Fig. 5.** Influence of each individual variable in the GAMs model on C$_2$H$_6$ troDMF from 2015 to 2020 over Hefei, eastern China. (*a*) to (*i*) are for zonal wind (*ua*), meridional wind (*va*), convection wind (*omega*), pressure (*pres*), tropopause height (*troph*), H$_2$O troDMF (*qv*), temperature (*temp*), CH$_4$ troDMF (*ch$_4$*), and CO troDMF (*co*), respectively. The DOFS of each smoothing function is included in the bracket in each panel. The *x*-axis represents variation range of each variable and the *y*-axis represents relative percentage change of C$_2$H$_6$ troDMF relative to its annual mean value.

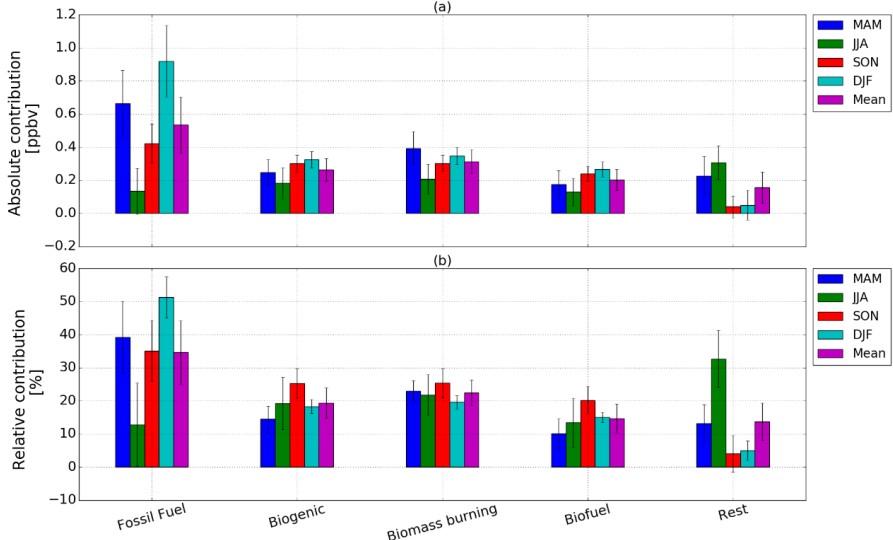

**Fig. 6.** Absolute (a) and relative (b) contributions of fossil fuel, biogenic, biomass burning, and biofuel emission sources to the observed C$_2$H$_6$ abundance from 2015 to 2020 over Hefei, eastern China. The remaining contribution calculated as the difference between the BASE simulation and the sum of anthropogenic and natural contributions is also shown. All contributions are grouped by season. Vertical error bars represent 1-σ standard variation.

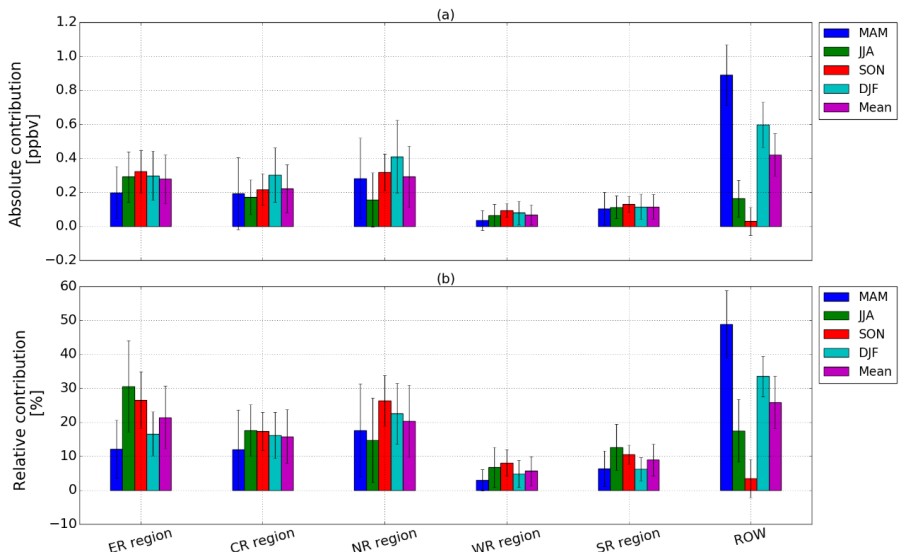

**Fig. 7.** Absolute (a) and relative (b) contributions of ER, CR, NR, WR, SR, and ROW regions to the observed $C_2H_6$
abundance from 2015 to 2020 over Hefei, eastern China. All contributions are grouped by season. Geographical
definition of each region is summarised in Table 3. Vertical error bars represent 1-σ standard variation.

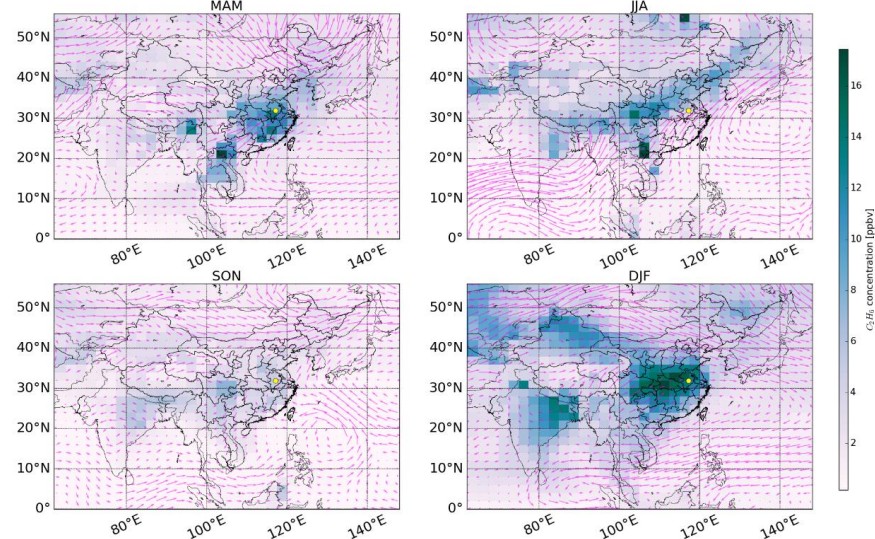

**Fig. 8.** Spatial distribution of $C_2H_6$ tropDMF in the GEOS-Chem BASE simulations in different seasons. The arrows
indicate horizontal wind vectors at 900 hPa; the observation site is marked with a yellow dot. Meteorological fields
are from the GEOS-FP 0.25° × 0.3125° dataset.



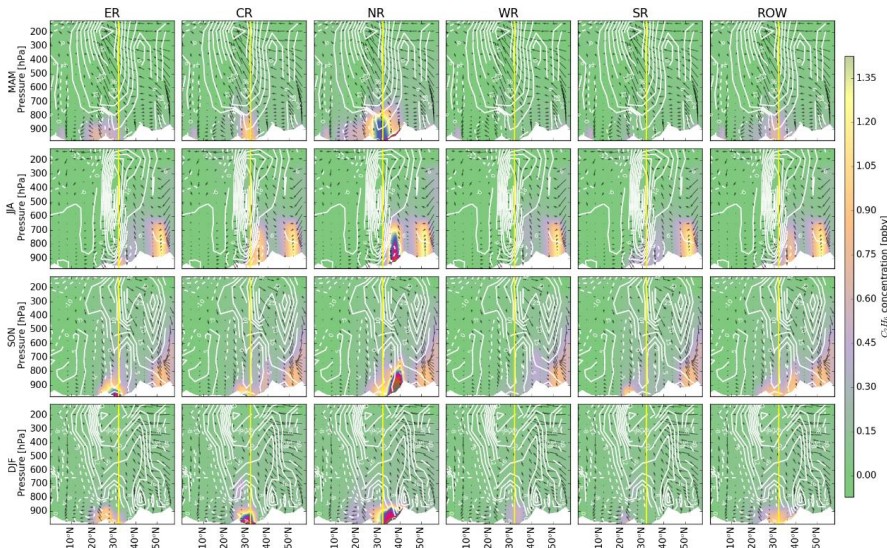

**Fig. 9.** The first row shows the latitude–height distributions of C₂H₆ VMR averaged over 113–121° E in spring
(MAM), originating in different source regions (corresponding to different columns). See Table 3 for latitude and
longitude definitions. The white area indicates topography, and the white contours at intervals of 6 m s⁻¹ are the
easterly (dashed) and westerly (solid) mean meridional winds; the wind vectors (consisting of zonal wind in m s⁻¹
and vertical velocity in units of Pa s⁻¹) are represented by arrows; the observation site is marked with a yellow line.
The second to fourth rows are the same as the first row but for summer (JJA), autumn (SON), and winter (DJF),
respectively. Meteorological fields are from the GEOS-FP 0.25° × 0.3125° dataset.
***Appendix.***

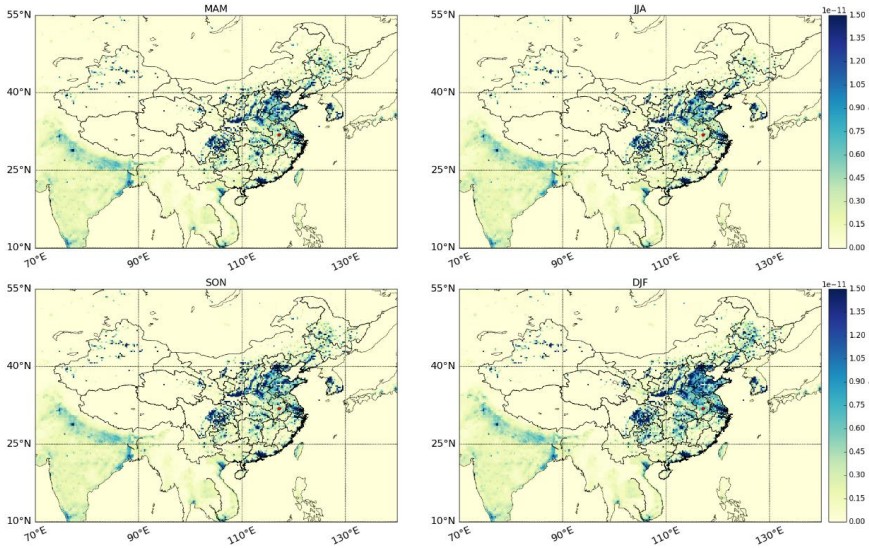

**Fig. A1**. C₂H₆ emission distribution in 2019 (0.25°×0.25°) from the Multi-resolution Emission Inventory for China
(MEIC) over China and suroundings. Units are in kgC/m²/s. The observation site is marked with a red dot.
