# Peer review of "Reduction in C2H6 from 2015 to 2020 over Hefei, eastern China points to air quality improvement in China"

_Atmospheric Chemistry and Physics, 2021_

## Author Comment (AC1)

**Response to Referee #1:**

Thanks very much for your comments, suggestions and recommendation with respect to improve this paper. The response to all your comments are listed below.

This paper presents a study to quantify the variability, source and transport of Ethane ($C_2H_6$) using solar absorption measurements from a ground-based high resolution Fourier transform infrared spectrometer performed over Hefei, eastern China. The measurements of five years (2015 - 2020) have been used to evaluate the GEOS-Chem model simulations, as well as to analyze dependencies on meteorological and emission factors using generalized additive models. Finally, the authors highlighted the sensitivities of model results to quantify relative contributions of various source categories and regions to the observed $C_2H_6$ abundances.

I have some concerns, which are mentioned below in the major and minor comments section. I recommend the publication of the manuscript after these points are addressed.

**Response:** All your comments listed below have been addressed. Please check the point by point response as follows.

**Major Comments**

1. P7 L23: It is hard to see the difference in the $C_2H_6$ emission distribution from the plots of Fig. A1. Perhaps, showing the relative difference w.r.t. the annual mean would be a good way to highlight the seasonal change.

**Response:** In the revised version, we have plotted both the absolute $C_2H_6$ emission distribution and the relative difference with respect to the annual mean value. Please check Fig. S1 for details.

2. P10 L4: the overestimation of 17.4% in December is not so evident from the figure. Please give more information on how this is calculated?

**Response:** This is a typing mistake. It is actually underestimation rather than overestimation. We have change it to adapt Figure 3. This information is calculated by (monthly mean of GEOS minus monthly mean of FTIR)/monthly mean of FTIR). We have inserted this equation in the revised version. By extending the time series of GEOS-Chem to match the FTIR observations, i.e., we run one year more for GEOS-Chem, now 17.4% becomes 14.6%. Please check section 3 for details.

3. P11 L40: the values do not match the figure, e.g., biogenic, please verify the absolute and relative contribution for other components as well.

**Response:** We have verified the absolute and relative contributions for all components in the revised version. Please check abstract, section 5.1, and conclusion for details.

4. P11 L41: these numbers should change based on the corrections done for the above comment.

**Response:** Done. Please check abstract, section 5.1, and conclusion for details.

5. Figure 3: Please provide some explanation on why the measurement uncertainties are lower during the summer time and vice-a-versa.

**Response:** It's a visual illusion but not real. In order to save time (for the huge computation task in this study), we did not run error analysis for all retrievals but only for a randomly selected $C_2H_6$ retrieval. This approximation did not affect any points of this study.

The retrieval errors added to all individual measurements in Fig.3 are 6.21% which is deduced from the error budget of randomly selected $C_2H_6$ retrieval at Hefei in Table 1. High levels of $C_2H_6$ troDMF occur in winter and low levels of $C_2H_6$ troDMF occur in summer. As a result, for a constant retrieval error (6.21%), the absolute uncertainties in summer are lower than those in winter. The observed $C_2H_6$ troDMF reached a minimum monthly mean value of (0.36 ± 0.26) ppbv in July and a maximum monthly mean value of (1.76 ± 0.35) ppbv in December. In this case, the monthly mean absolute uncertainties of $C_2H_6$ in summer and winter are 0.022 (0.36*6.21%) and 0.11 ppbv (1.76*6.21%), respectively. The former value is lower than the later one.

We have included a clarification to avoid this misleading, i.e., "A retrieval error of 6.21% in Table 1 was used to estimate the retrieval uncertainties of all observations. As a result, the uncertainties in winter are larger than those in summer due to a higher $C_2H_6$ level in the season". Please check the caption of Figure 3 for details.

**Minor Comments**:
1. P1 L32: together with atmospheric modelling
**Response:** Done.

2. P1 L36: no brackets needed
**Response:** Done.

3. P3 L4: ... namely the infrared working group (IRWG) of the Network for the Detection of ...
**Response:** Done.

4. P4 L19: please provide the spectral range of the NIR and MIR observations.
**Response:** Done. "This FTIR observatory alternately saved near infrared (NIR) and middle infrared (MIR) solar spectra in routine observations, with spectral ranges of 4,000 to 11,000 cm$^{-1}$ and 500 to 8,500 cm$^{-1}$, respectively."

5. P5 L30: shouldn't this be table 1?

**Response:** It is table 3 indeed but to avoid misleading, we have removed the words "and Table 3" since Fig.2 already conveys this message.

6. P6 L7: zero level uncertainty reported is different from the value in table 1

**Response:** It should be 1.45% and we have modified it to be consistent with Table1.

7. P14 L36: and -3.93%, respectively

**Response:** Done.

---

## Author Comment (AC2)

**Response to Referee #2**:

Thanks very much for your comments, suggestions and recommendation with respect to improve this paper. The response to all your comments are listed below.

This study by Sun et al., for the first time, presents and quantifies the variability, source, and transport of $C_2H_6$ over densely populated and industrialized eastern China by using ground-based high resolution FTIR observation, GEOS-Chem model simulation, and the analysis of meteorological fields. The dependencies of $C_2H_6$ on meteorological factors and co-emitted gases are also analyzed by using generalized additive models (GAMs). The ground-based FTIR $C_2H_6$ time series are applied to evaluate the GEOS-Chem model for the specifics of $C_2H_6$ simulation over eastern China. The authors further run a series of GEOS-Chem sensitivity simulations to quantify relative contributions of various source categories and regions to the observed $C_2H_6$ abundance. They also conclude that there is a decreasing change rate in $C_2H_6$ due to the decrease in local and transported $C_2H_6$ emissions, which points to air quality improvement in China in recent years. The three dimensional (3D) transport inflow and outflow pathways of $C_2H_6$ over the observation site are finally determined by the GEOS-Chem sensitivity simulations and the analysis of meteorological fields. Overall, this manuscript is well written, structured, and its topic fits well in the scope of ACP. I believe that the results of this study could be of interest to the general atmospheric science community and should be in the literature. However, a couple of minor points that should be corrected/clarified before publication.

**Response:** All your comments listed below have been addressed. Please check the point by point response as follows.

**Specific comments:**

1. GEOS-Chem is a powerful tool for source attribution of atmospheric composition; however, I feel that the way the authors implemented the model raises misleading and should be clarified. In Section 2.2, the GEOS-Chem model setup is described. On L25-26, the authors state a 1hr time step for surface variables and boundary layer heights. I am not sure what surface variables are in this case or is the boundary layer time step? I am guessing these are the emissions and boundary layer mixing time steps? Additionally, given the importance of the boundary layer in this studies, the authors should state which boundary layer mixing scheme was used. The authors also state a 3hr time step of all other variables. Is this referring to transport and chemistry time steps? If so, this seems exceedingly long, especially for the full-chemistry simulation. All these should be clearly described or clarified.

**Response:** We apologize for the confusion. The original text in L25-26 in section 2.2

are for time step of the input meteorological fields from GEOS-FP, i.e. 1-hour for surface meteorological variables (e.g. surface temperature) and PBL heights and 3-hour for other meteorological variables. This is not the time step for chemistry, emission, and transport. We now state in the section 2.2 to avoid confusion: "The time step used in the model are 10 minutes for transport and 20 minutes for chemistry and emissions, as recommend for the GEOS-Chem full-chemistry simulation at $2 \times 2.5$ (Philip et al., 2016)."

The time step of PBL mixing follows that of transport (i.e. 10 minutes). We add the following text for the PBL scheme description: "The non-local scheme for the boundary layer mixing process are described in Lin and McElroy (2010)."
Lin J. T., Mcelroy M. B.: Impacts of boundary layer mixing on pollutant vertical profiles in the lower troposphere: Implications to satellite remote sensing. Atmos. Environ., 2010, 44(14):1726-1739.

2. I noticed in Figure 3 that there is a mismatch in terms of time coverage between GEOS-Chem and FTIR observations. The time series of GEOS-Chem is about one year less than the FTIR observations. Is it possible to extend the time series of GEOS-Chem to match the FTIR observations?
**Response:** We have extended the time series of GEOS-Chem to match the FTIR observations. Please see Figure 3 in the revised version for details.

**Technical comments:**
1. Please provide correlation and error budget figures for the validations of GEOS-Chem model and GAMs model. I suggest it can be added to the supplement.
**Response:** We have included these figures in the supplement. Correlation plots for the GEOS-Chem−to−FTIR data pairs from 2015 to 2020 over Hefei are shown in Fig. S2. Correlation plots for the GAMs model−to−FTIR data pairs from 2015 to 2020 over Hefei are shown in Fig. S3.

2. Figure 1: Please add (a), (b) and (c) to 3 subplots, and explained it in the caption.
**Response:** We have included (a), (b) and (c) into 3 subplots and explained them in the caption. Please see Figure 1 for details.

3. Figure 3: short data gaps of up to a few months have occurred between 2016 and 2017. Please explain the reason. Is this due to data quality control?
**Response:** This is due to instrument problem, we have clarified this in the revised paper, "The instrument has been operating continuously since its installation; however, short data gaps of up to eight months have occurred due to a scanner problem between November 2016 and July 2017". Please see section 2.1 for details.

4. The atmospheric circulation pattern technique mentioned in this study is actually the analysis of the meteorological fields. So for clarity, please replace all atmospheric circulation pattern technique terms with the analysis of the meteorological fields.

**Response:** We have replaced all "atmospheric circulation pattern technique" terms with the "analysis of the meteorological fields"

5. With respect to language, the text is in my impression occasionally penetrated with incorrect/awkward phrases. For example, from my perspective, "Conclusions" rather than "Summary and conclusion" is sufficient for the title of section 6. I am not a native speaker, therefore I did not attempt to correct all these flaws throughout the whole paper. Instead, I would recommend a linguistic revision of the whole text: I assume that either one of the coauthors with a good command of the English language or ACP can provide support for this task.

**Response:** First, "Summary and conclusion" has been revised to "Conclusions". In addition, one of the coauthors with good command of English has copy-edited the rest parts. I assume that the copy editing phase will further improve the grammars. Since all revisions are minor. We did not marked up the changes in the revised paper.